# The Association between Aortic Calcification Index and Urinary Stones: A Cross-Sectional Study

**DOI:** 10.3390/jcm11195884

**Published:** 2022-10-05

**Authors:** Weinan Chen, Liulin Xiong, Qingquan Xu, Liang Chen, Xiaobo Huang

**Affiliations:** Department of Urology, Peking University People’s Hospital, Beijing 100044, China

**Keywords:** urinary stones, vascular calcification, ACI, risk factor

## Abstract

Background: It is believed that vascular calcification and urinary stones may possibly have a shared mechanism. However, the association between vascular calcification and urinary stones is largely unexplored. Using the aortic calcification index (ACI) as a clinical indicator of vascular calcification, the present study aimed to examine the association between the ACI and urinary stones. Methods: This cross-sectional study included 282 patients hospitalized for either urinary stones or other urological diseases from January 2020 to December 2021 at the Department of Urology and Lithotripsy in Peking University People’s Hospital. Among them, 137 and 145 patients were divided into the stone group and the non-stone group. Multivariable logistic regression analysis was performed to examine the association between the ACI and urinary stones. The restricted cubic splines model was used to further explore the dose–response relationship between the ACI and urinary stones. Results: The median (Q1–Q3) age of the study population was 59.0 (47.0–67.0) years. After adjusting coronary heart disease, triglycerides, glucose, serum creatinine, uric acid, urea, calcium, and eGFR, the ACI was independently associated with urinary stones (odds ratio [OR], 1.07; 95% confidence interval [CI], 1.03–1.11) in a linear dose–response pattern (*p* for non-linearity =0.153). Age was found to interact with the effect of the ACI on urinary stones (*p* for interaction <0.001). Conclusions: This study found that the ACI was independently associated with urinary stones in a linear dose–response manner. Our results indicate that the ACI might be a helpful indicator for identifying populations at risk for urinary stones.

## 1. Introduction

Urolithiasis is one of the most common urological diseases worldwide. In Asia, the prevalence of urolithiasis is estimated to be 1% to 5%, while in Western countries this figure is approximately 2–3 times higher [1,2]. Due to the high rates of new and recurrent stones, as well as the consistently increasing prevalence worldwide [3], it is imperative to identify the risk population and address obstacles in the prevention of urinary stones.

Among all the risk factors of urinary stones, metabolic abnormalities and cardiovascular diseases are considered to play important roles in urinary stones formation. Patients with urinary stones are often found to have type 2 diabetes, atherosclerosis, coronary heart disease, and other cardiovascular diseases [4,5,6,7,8,9]. As vascular calcification is closely related to metabolic abnormalities and many cardiovascular diseases [10,11], the association of increased vascular calcification with stones are of marked interest in recent years. An important indicator to assess the degree of aortic calcification is the aortic calcification index (ACI). Studies on whether individuals with a higher ACI are associated with an increased prevalence in urinary stones may hint at a link between the two conditions. However, to date this topic is largely unexplored.

Therefore, we conducted a cross-sectional study to investigate the association between the ACI and urinary stones in a group of patients hospitalized for urinary stones or other urological diseases in our hospital.

## 2. Materials and Methods

### 2.1. Study Population

All participants in this study were patients at the Department of Urology and Lithotripsy in Peking University People’s Hospital in Beijing, China. Initially, we included patients hospitalized for either urinary stones (patients hospitalized for recurrent urinary stones were excluded at the beginning since previous treatment of urinary stones may affect blood biochemical indicators or vascular calcification process) or other urological diseases (mainly including renal cyst, renal hamartoma larger than 4 cm, and benign prostatic hyperplasia, etc.) from 1 January 2020 to 31 December 2021 (*n* = 311). Patients with bilateral ureteral lesions such as bilateral ureteral calculi were initially excluded from the study population since their obstructive symptoms may affect creatinine and estimated glomerular filtration rate (eGFR) levels, which were important covariates in this study.

Patients who aged <18 or ≥90 years (*n* = 2), did not undergo abdominal computed tomography examination (*n* = 22), had missing values for laboratory tests (*n* = 4), or had a history of cancer (*n* = 1) were excluded from the study. Finally, 282 patients were included in the analyses (Figure 1). Among them, 137 were divided into the stone group, while 145 were in the non-stone group.

This study was approved by the Institutional Review Board of Peking University People’s Hospital and has therefore been performed in accordance with the ethical standards as laid down in the 1964 Declaration of Helsinki and its later amendments or comparable ethical standards.

### 2.2. Data Collection

Data on demographic characteristics and medical history of each participant were collected at the date of their hospitalization. Height and weight were measured using calibrated instruments with standard protocols and recorded by trained nurses. Fasting venous blood samples were collected from each participant as part of routine clinical examination. Total cholesterol, triglycerides, glucose, creatinine, uric acid, urea, calcium, phosphate, potassium, sodium, chloride, and urine pH were measured using calibrated automatic analyzers.

### 2.3. Assessment of ACI

The ACI was quantitatively measured using an abdominal computed tomography examination by evaluating 10 slices of the aorta scanned at 10-mm intervals above the bifurcation of the common iliac arteries. Each slice was divided into 12 sectors, and the numbers of sectors with calcification were counted. For example, if 4 out of 12 sectors were calcified in slice 1, this was scored as 4/12 = 33.3%. The ACI (%) was calculated by averaging the percentage of calcification-positive sectors in slices 1 to 10. The ACI was assessed by a single investigator in a blinded manner. Each participant underwent an abdominal computed tomographic examination at the date of hospitalization.

### 2.4. Diagnosis of Urinary Stones

An abdominal computed tomography examination was performed for all patients to determine the presence of urinary stones. The examination was performed using calibrated instruments with standard protocols and recorded by specialists in the Department of Radiology in our hospital. Each participant underwent the abdominal computed tomography examination at the date of hospitalization.

### 2.5. Assessment of Covariates

Body mass index (BMI) was calculated as body weight in kilograms divided by the square of height in meters. Age, BMI, serum analytes (including triglycerides, creatinine, uric acid, etc.), and urine pH were analyzed as continuous variables. Sex was categorized as male or female. Participants were identified as normal weight (BMI < 28 kg/m^2^) or obesity (BMI ≥ 28 kg/m^2^) according to the cut-off recommended by the Working Group on Obesity in China [12]. The estimated (eGFR) was calculated using the Chronic Kidney Disease Epidemiology Collaboration (CKD-EPI) equation [13]. Chronic kidney disease (CKD) stage 3–5 was identified if the eGFR < 60 mL/min per 1.73 m^2^. Dyslipidemia was defined as having triglyceride ≥1.7 mmol/L or total cholesterol ≥5.2 mmol/L or using lipid-lowering medication. Diabetes was defined as fasting blood glucose ≥6.1 mmol/L or self-reported diabetes or using antidiabetic drugs. Coronary health disease (CHD) was based on participants’ self-reported diagnosis of CHD.

### 2.6. Statistical Analyses

Continuous variables with normal or skewed distributions are described as the mean ±standard deviation (SD) or median (Q1-Q3) and were compared using the *t*-test or Mann–Whitney U-test. Categorical variables are presented as frequencies and percentages and were compared using the χ^2^ test. The optimal cut-off value of the ACI for urinary stones was calculated with the receiver operating characteristic curve. Univariable and multivariable logistic regression analyses were performed to identify the significant factors associated with urinary stones. Factors with a *p*-value < 0.05 in the univariable logistic regression model were included into the multivariable logistic regression model to examine the independent risk factors of urinary stones. Since there may be collinearity between diabetes, dyslipidemia and glucose and lipid indicators, we included only continuous variables (glucose and lipid indicators) in the multivariable model to improve the model’s goodness of fit. Restricted cubic splines with knots at the 5th, 27.5th, 50th, 72.5th, and 95th percentiles were used to further explore the shape of the dose–response relationship between ACI and urinary stones.

The multivariable-adjusted model was stratified by baseline demographic and clinic characteristics as potential modifiers: age (<60 or ≥60 years), sex (male or female), and obesity (no or yes). The multiplicative interaction terms between these subgroups and weight/WC change were added to the fully adjusted model, and models with and without multiplicative interaction terms were compared using the likelihood-ratio test.

All statistical analyses were performed using R version 3.6.2. The “glm” was used to conduct logistic regression analyses. The “pROC” package was used to draw the receiver operating characteristic curve. The “rms” and “ggplot2” packages were used to perform the restricted cubic splines analysis and draw the boxplot and bar plot between groups. All *p*-values were two-sided, and statistical significance was defined as *p* < 0.05.

## 3. Results

### 3.1. Characteristics of the Study Population

Characteristics of the study population between non-stone and stone groups are presented in Table 1. Among the 282 patients, 68.8% were male, the median (Q1–Q3) age was 59.0 (47.0–67.0) years, the mean ± SD BMI was 25.12 ± 3.56 kg/m^2^. 137 (48.6%) patients belonged to the stone group. Compared to the non-stone group, the prevalence of dyslipidemia (56.9% vs. 36.6%, *p* = 0.001), CHD (8.0% vs. 2.1%, *p* = 0.042), and CKD stage 3–5 (16.8% vs. 6.2%, *p* = 0.009) were significantly higher in the stone group. In addition, triglycerides, creatinine, uric acid, Ca, and ACI were significantly increased in the stone group compared to their counterparts. However, phosphoric acid was significantly higher in the non-stone group. No other significant differences were observed between the two groups.

### 3.2. ACI and the Prevalence of Urinary Stones

The median ACI in all participants was 7.69% (Table 1, Figure 2). Participants in the stone group had a significantly higher ACI than those in the non-stone group (10.45% vs. 6.44%, *p* < 0.001; Table 1). The optimal cut-off value of the ACI for urinary stones was 9.29%. Patients with an ACI ≥ 9.29% had a significantly higher prevalence of urinary stones than those with an ACI < 9.29% (65.5% vs. 36.2%, *p* < 0.001).

### 3.3. Association between ACI and Urinary Stones

In the univariable analyses, dyslipidemia, CHD, CKD stage 3–5, glucose, triglycerides, eGFR, creatinine, uric acid, urea, Ca, and the ACI were significantly associated with urinary stones (Table 2). All these factors were positively associated with urinary stones, except for eGFR. In the multivariable analyses, after adjusting CHD, triglycerides, glucose, serum creatinine, uric acid, urea, calcium, and eGFR, the ACI was still independently associated with urinary stones (odds ratio [OR], 1.07; 95% confidence interval [CI], 1.03–1.11; *p* < 0.001; Table 2). In the restricted cubic splines analyses, the ACI was found to have a linear dose–response relationship with urinary stones, with *p* for non-linearity =0.153 (Figure 3).

Association was obtained through restricted cubic splines analyses adjusted for CHD, triglycerides, glucose, serum creatinine, uric acid, urea, calcium, and eGFR. Reference values of ACI =9.29%. The solid lines and shaded areas represent the OR and corresponding 95% CI.

### 3.4. Subgroup Analyses

In subgroup analyses (Figure 4), the associations between the ACI and urinary stones did not differ by sex and obesity status (all *p* for interaction >0.05). However, an interaction was observed in the association of the ACI with urinary stones across age subgroups (*p* for interaction <0.001). The association between the ACI and urinary stones seemed to be stronger among individuals aged <60 years (OR, 1.25; 95% CI, 1.13–1.40), compared to their elderly counterparts (OR, 1.03; 95% CI, 0.99–1.08).

The models were adjusted for CHD, triglycerides, glucose, serum creatinine, uric acid, urea, calcium, and eGFR.

## 4. Discussion

Urinary stones are known to be one of the adverse outcomes of cardiovascular diseases (e.g., atherosclerosis, CHD, etc.) and metabolic syndrome [3,7,9], whereas few studies have investigated the effect of vascular calcification on the formation of urinary stones. In the present study, using the ACI as the indicator of vascular calcification, we found that the ACI was significantly associated with urinary stones in a linear dose–response manner. These findings indicate that ACI is an independent risk factor for stones formation, and the ACI might be a helpful indicator for identifying populations at risk for urinary stones.

Previous clinical studies on the association between vascular calcification and urinary stones are few, and the results of these studies are inconsistent [14,15,16,17,18,19]. A study conducted in a health screening population found that the stone group had a significantly higher coronary artery calcium score than the non-stone group, and multivariable analyses showed that the coronary artery calcium score ratio was associated with nephrolithiasis (OR, 1.31; 95% CI: 1.00–1.71) [14]. The Multi-Ethnic Study of Atherosclerosis (MESA) found that the coronary artery calcium score was a significant risk factor for recurrent stones (OR, 1.80; 95% CI, 1.22–2.67) but not for a single stone (OR, 0.95; 95% CI, 0.64–1.41) [15]. A study involving 394 patients found that patients with mild arterial calcifications did not have a significantly higher risk for kidney stones, and that those with intermediate or severe arterial calcifications were 1.9 times more likely to have stones than their counterparts [16]. In addition, two studies exploring the association between abdominal aortic calcification and calcium oxalate calculi demonstrated that abdominal aortic calcification was a risk factor for calcium oxalate calculi, with ORs and 95% CIs of 1.35 (1.00–1.82) and 1.25 (1.00–1.56), respectively [16,17,18]. Contrary to these findings, a hospital-based study did not find any association between abdominal aortic calcification and urolithiasis [19]. Our results show that individuals with urinary stones had a significantly higher ACI than those in the non-stone group, and the prevalence of urinary stones in individuals with a higher ACI was significantly higher compared to those with a relatively lower ACI. Moreover, multivariable analyses showed that the ACI was an independent risk factor for urinary stones. To the best of our knowledge, this study is the first to explore the dose–response relationship between the ACI and urinary stones. All these results demonstrate that the ACI has a significant effect on urinary stones. These findings provide additional support for the hypothesis that there might be a shared mechanism between vascular calcification and urinary stones.

Growing evidence has linked urolithiasis to various metabolic diseases, including metabolic syndrome, diabetes, and cardiovascular events [3,5,6,7]. Consistent with previous evidence, we also found that the prevalence of CHD and dyslipidemia was significantly higher among individuals with urinary stones. These systemic diseases are closely related to vascular calcification. Although a shared origin of urinary stone formation and vascular calcification is biologically plausible, the underlying mechanism remains uncertain. A decrease in calcification inhibitors (e. g. fetuin-A, matrix Gla protein, etc.) may be the link between vascular calcification and nephrolithiasis. Studies have found that vascular calcification and kidney stones are both related to low serum levels of fetuin-A [19,20,21]. In addition, the dysfunction in the calcium-sensing receptor can lead to a dysregulation in renal calcium handling and the failure to prevent vascular calcification [22]. Furthermore, matrix vesicles, which are the extracellular membrane-bound vesicles and participate in the progression of cardiovascular calcification, could cause nucleation and crystallization, lead to crystal deposition in the renal papilla, and eventually lead to the formation of stones [23]. Finally, evidence has shown that in kidney stone formers, abdominal aortic calcifications was generally accompanied by low urinary pH and hypocitraturia, suggesting vascular calcifications may be involved in the lithogenic process [24].

Interestingly, a stronger association between the ACI and urinary stones was observed in participants younger than 60 years compared to their elderly counterparts in this study. A possible explanation is that compared to younger participants, a large proportion of urinary stones in elderly participants may relate to a high prevalence of metabolic diseases, urinary tract infection, low level of physical activity [23,24,25,26], etc., and the ACI added only modestly deleterious effects on the relative scale. Such findings somehow suggest that high levels of the ACI in young adults should be of concern in urologic practice. Nevertheless, further studies are needed to elucidate the mechanisms of the age difference in the association between the ACI and urinary stones.

The strengths of this study include strict adherence to the measurement procedure and study protocol, an exploration of the dose–response relationship between the ACI and urinary stones, and further analyses in various subgroups. However, several limitations in this study need to be noted. First, since this study adopted a cross-sectional design, the association between the ACI and a risk for urinary stones could not be evaluated. Therefore, longitudinal studies with a large sample size are needed to further examine the association between vascular calcification and urinary stones. Second, there may be a possible selection bias in the study population because the non-stone group consisted of patients admitted to our hospital for other urological diseases rather than healthy individuals free from urinary stones. The prevalence of patients with a higher ACI may be overestimated since the study population consisted of patients admitted to hospital. Therefore, because our study population may not be a good representative sample of the general population or patients in general urology centers, the results of this study should be carefully generalized to other populations. Third, although we adjusted for various confounders in the multivariable analyses, we cannot exclude the possibility of residual confounding due to unmeasured variables, such as dietary patterns, lifestyle information, and hypertension. Fourth, the differences on the association between the ACI and various types of stones may provide additional information on the potential mechanism between vascular calcification and urinary stones. Unfortunately, because the stone compositions of a proportion of the patients were not assessed, we were unable to conduct further analyses on this topic.

## 5. Conclusions

The present study found that the ACI was independently associated with urinary stones in a linear dose–response manner. The association was evident across sex, BMI, and age subgroups, with age interacting with the effect of the ACI on urinary stones. Our results indicate that the ACI might be a helpful indicator for identifying populations at risk for urinary stones.

## Figures and Tables

**Figure 1 jcm-11-05884-f001:**
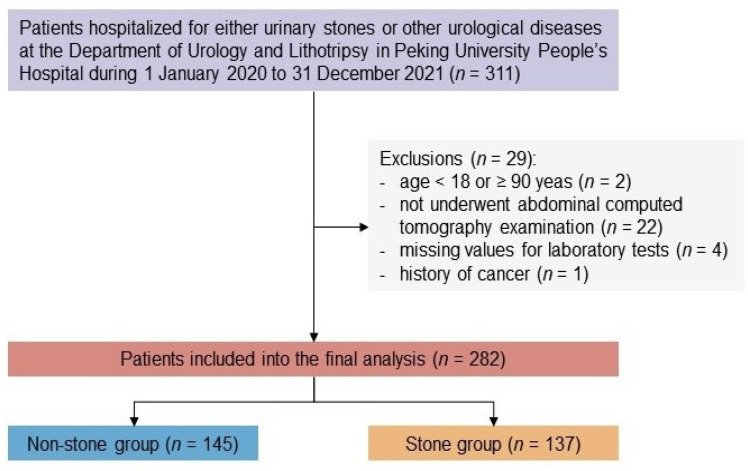
Flowchart of the study population.

**Figure 2 jcm-11-05884-f002:**
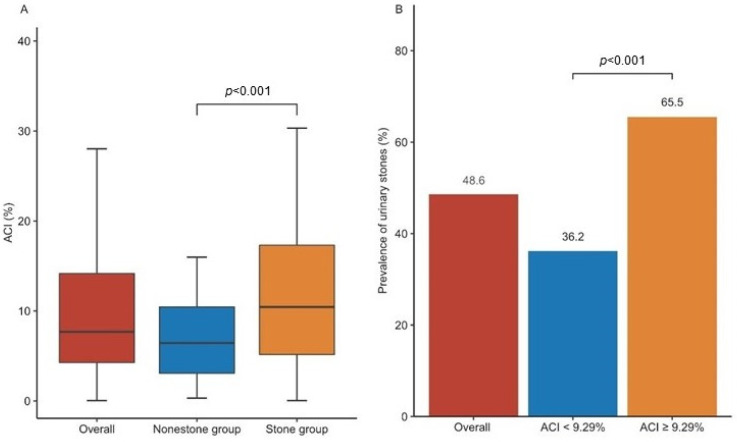
Association between aortic calcification index (ACI) and urinary stones. (**A**) Comparison of ACI in the non-stone group and the stone group; (**B**) The prevalence of urinary stones in patients with ACI < 9.29% and patients with ACI ≥ 9.29%.

**Figure 3 jcm-11-05884-f003:**
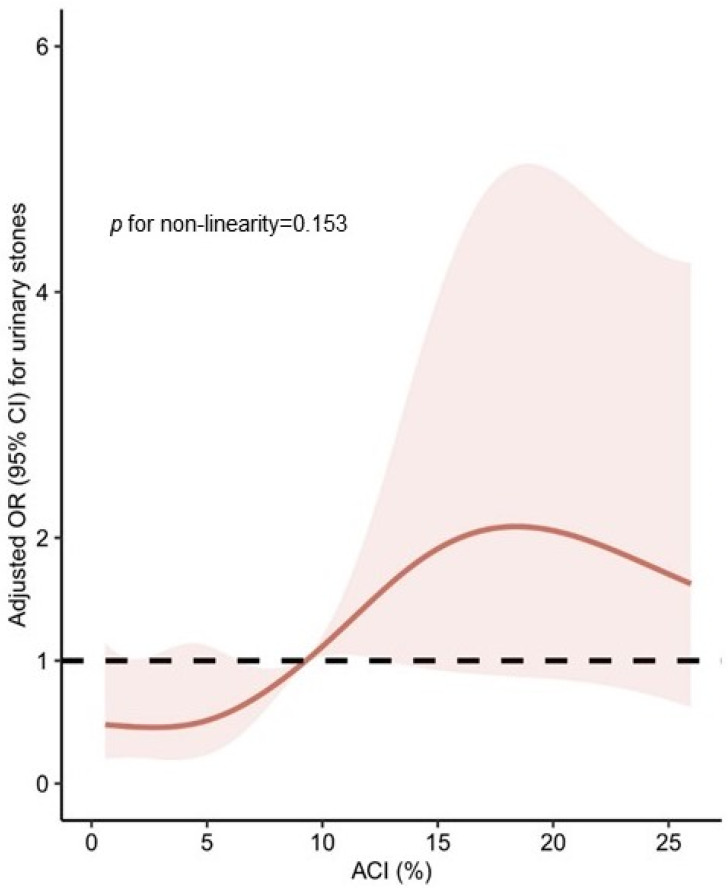
Dose–response relationship between aortic calcification index (ACI) and urinary stones.

**Figure 4 jcm-11-05884-f004:**
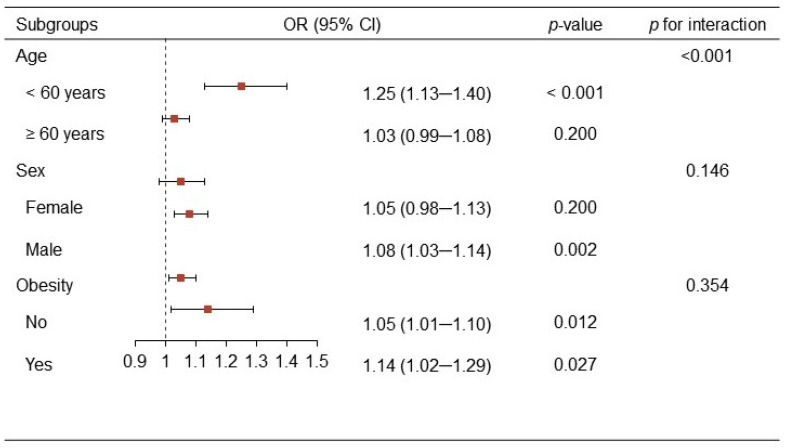
Subgroup analyses of the association between aortic calcification index (ACI) and urinary stones by age (<60 years or ≥60 years), sex (female or male) and by obesity (no or yes).

**Table 1 jcm-11-05884-t001:** Characteristics of the study population.

Characteristics	Overall (*n* = 282)	Non-Stone Group (*n* = 145)	Stone Group (*n* = 137)	*p*-Value
Male, *n* (%)	194 (68.8)	100 (69.0)	94 (68.6)	1.000
Age, years ^a^	59.0 (47.0, 67.0)	62.0 (48.0, 69.0)	57.0 (47.0, 66.0)	0.080
BMI, kg/m^2 b^	25.12 ± 3.56	24.98 ± 3.55	25.27 ± 3.58	0.493
Obesity, *n* (%)	57 (20.2)	28 (19.3)	29 (21.2)	0.810
Diabetes, *n* (%)	88 (31.2)	38 (26.2)	50 (36.5)	0.083
Dyslipidemia, *n* (%)	131 (46.5)	53 (36.6)	78 (56.9)	0.001
CHD, *n* (%)	14 (5.0)	3 (2.1)	11 (8.0)	0.042
CKD stage 3–5, *n* (%)	32 (11.3)	9 (6.2)	23 (16.8)	0.009
Glucose, mmol/L ^a^	5.56 (4.99, 6.68)	5.45 (5.01, 6.36)	5.79 (4.98, 6.91)	0.221
Total cholesterol, mmol/L ^a^	4.34 (3.73, 4.97)	4.30 (3.67, 4.92)	4.37 (3.74, 5.21)	0.355
Triglycerides, mmol/L ^a^	1.39 (1.02, 2.00)	1.27 (0.94, 1.76)	1.58 (1.13, 2.26)	<0.001
eGFR, ml/min per 1.73 m^2 a^	92.32 (76.37, 101.82)	92.50 (81.04, 103.43)	91.96 (71.12, 100.72)	0.081
Serum creatinine, μmol/L ^a^	75.00 (65.00, 89.00)	73.00 (64.00, 84.00)	77.00 (66.00, 96.00)	0.012
Uric acid, μmol/L ^a^	336.50 (283.00, 404.00)	313.00 (267.00, 394.00)	352.00 (303.00, 410.00)	0.007
Urea, mmol/L ^a^	5.18 (4.34, 6.08)	5.10 (4.27, 5.95)	5.31 (4.40, 6.47)	0.102
Calcium, mmol/L ^a^	2.33 (2.25, 2.40)	2.32 (2.25, 2.38)	2.34 (2.27, 2.43)	0.046
Phosphate, mmol/L ^a^	1.16 (1.05, 1.26)	1.19 (1.08, 1.27)	1.12 (1.00, 1.25)	0.008
Potassium, mmol/L ^b^	3.95 ± 0.37	3.99 ± 0.34	3.91 ± 0.40	0.070
Sodium, mmol/L ^a^	139.00 (138.00, 140.00)	139.00 (137.00, 141.00)	139.00 (138.00, 140.00)	0.534
Chloride, mmol/L ^a^	104.00 (103.00, 106.00)	104.00 (103.00, 107.00)	104.00 (103.00, 106.00)	0.789
Urine pH ^a^	6.00 (5.50, 6.50)	6.00 (5.50, 6.50)	6.00 (5.50, 6.50)	0.713
ACI (%) ^a^	7.69 (4.26, 14.17)	6.44 (3.08, 10.45)	10.45 (5.16, 17.31)	<0.001

^a^ Data are presented as median (Q1–Q3); ^b^ Data are presented as mean ± SD; Abbreviations: BMI, body mass index; CHD, coronary heart disease; CKD, chronic kidney disease; eGFR, estimated glomerular filtration rate; ACI, aortic calcification index.

**Table 2 jcm-11-05884-t002:** Univariable and multivariable analyses of the risk factors of urinary stones.

Factors	Univariable Analyses		Multivariable Analyses	
	Odds Ratio (95% CI)	*p*-Value	Odds Ratio (95% CI) ^a^	*p*-Value
Male	0.98 (0.59–1.63)	0.949		
Age	0.99 (0.97–1.01)	0.224		
BMI	1.02 (0.96–1.09)	0.492		
Obesity	1.12 (0.63–2.01)	0.698		
Diabetes	1.62 (0.97–2.69)	0.063		
Dyslipidemia	2.29 (1.42–3.70)	0.001		
CHD	4.13 (1.13–15.15)	0.032		
CKD stage 3–5	3.05 (1.36–6.85)	0.007		
Glucose	1.22 (1.06–1.41)	0.006	1.16 (1.00–1.35)	0.049
Total cholesterol	1.21 (0.95–1.53)	0.115		
Triglycerides	1.28 (1.02–1.61)	0.033		
eGFR	0.98 (0.97–1.00)	0.008	1.02 (1.00–1.05)	0.063
Serum creatinine	1.02 (1.01–1.03)	0.001	1.04 (1.02–1.06)	0.001
Uric acid	1.00 (1.00–1.01)	0.017		
Urea	1.18 (1.03–1.35)	0.016		
Calcium	6.19 (1.02–37.47)	0.047	5.92 (0.76–46.41)	0.091
Phosphate	0.49 (0.16–1.47)	0.204		
Potassium	0.55 (0.29–1.05)	0.071		
Sodium	0.99 (0.92–1.06)	0.710		
Chloride	1.01 (0.98–1.03)	0.639		
Urine pH	0.90 (0.66–1.24)	0.521		
ACI	1.06 (1.02–1.09)	< 0.001	1.07 (1.03–1.11)	<0.001

^a^ Multivariate logistic regression model was adjusted for ACI, CHD, Tg, glucose, creatinine, uric acid, urea, Ca, and eGFR. Abbreviations: BMI, body mass index; CHD, coronary heart disease; CKD, chronic kidney disease; eGFR, estimated glomerular filtration rate; ACI, aortic calcification index.

## Data Availability

Datasets used and/or analyzed during the current study are available from the corresponding author on reasonable request.

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
