# Peer review of "The Association between Aortic Calcification Index and Urinary Stones: A Cross-Sectional Study"

_jcm, 2022, doi:10.3390/jcm11195884_

Round 1

Reviewer 1 Report

This paper is exceptionally written. I don't remember the last paper I wrote with so few grammatical edits. I only have two small grammatical edits: 1) Intro, second paragraph "patients with urinary stones WERE often found" should be "ARE often found". Discussion, 1st sentence - Urinary stones "ARE KNOWN" instead of "were proved"

My big issues are with the methods and the fact that this topic has already been explored. The paper reads so well, it is hard for me to not recommend it. However, I'm not sure what new content this paper brings to the literature. You are missing a big citation (Stern et al, doi: 10.1089/end.2019.0243) which looked specifically at aortic calcifications and the risk of kidney stones. Patel et al took it further and looked at urinary abnormalities in stone formers with aortic calcifications (doi: 10.1089/end.2017.0350). How can your paper be re-written to bring something new to the field?

Other limitations - why ultrasound patients for stones when you have a CT scan? You can't include creatinine or GFR in the analysis if patients are hospitalized for stones and therefore potentially obstructed. Were all patients confirmed to not have obstructed renal systems when the labs were obtained? What indications for hospitalization were included? Inherently you would think that patients requiring hospitalization, stone patients or not, would be more unhealthy and therefore likely to have higher aortic calcifications. 

Reviewer 2 Report

- Dear colleagues, in your study population of 317 patients almost every patient (only 22 excluded) got a computed tomography. This does not represent the patient population in most centres I am sure. Are you already specialised in urolithiasis patients? Or did you only include patients from a urolithiasis subclinic? This might be a bias esp. for the non-stone group.

- Also recurrent stone formers (only 3?) were excluded. Can you elaborate why?

- What was the purpose of hospitalisation of the non-stone group (tumors + children excluded)? Mainly infection/trauma/LUTS?

- You mention in your methods that diagnosis of stone disease was done by ultrasound. Why not by CT if every patient included got one?

Round 2

Reviewer 1 Report

Nice edits. Again - well written. Unfortunately doesn't bring anything novel to the area but a nice manuscript to confirm the association.